# Larazotide Acetate Protects the Intestinal Mucosal Barrier from Anoxia/Reoxygenation Injury via Various Cellular Mechanisms

**DOI:** 10.3390/biomedicines13102483

**Published:** 2025-10-12

**Authors:** Jain Kim, Jay P. Madan, Sandeep Laumas, B. Radha Krishnan, Younggeon Jin

**Affiliations:** 1Department of Animal and Avian Sciences, University of Maryland, College Park, MD 20742, USA; 2Innovate Biopharmaceuticals Inc., Raleigh, NC 27615, USA

**Keywords:** larazotide acetate, tight junctions, intestinal barrier function, anoxia/reoxygenation injury

## Abstract

**Background/Objective**: Larazotide acetate (LA) is a synthetic octapeptide under development as a therapeutic candidate for celiac disease, acting to reduce intestinal permeability and regulate tight junctions (TJs). Although several studies have shown barrier-protective effects, the cellular mechanisms underlying LA’s actions in the intestinal epithelium remain unclear. This study aimed to elucidate the mechanistic roles of LA in maintaining intestinal epithelial integrity during cellular injury. **Methods:** C2BBe1 and leaky IPEC-J2 cell monolayers were pretreated with 10 mM LA and subjected to anoxia/reoxygenation (A/R) injury. Transepithelial electrical resistance (TEER), TJ protein localization, and phosphorylation of myosin light chain-2 (MLC-2) were analyzed. In addition, RNA sequencing was conducted to identify differentially expressed genes and signaling pathways affected by LA treatment. **Results:** LA pretreatment significantly increased TEER and preserved TJ protein organization during A/R injury. Transcriptomic analysis revealed enrichment of genes related to barrier regulation, small GTPase signaling, protein phosphorylation, proliferation, and migration. LA pretreatment markedly reduced MLC-2 phosphorylation, likely through modulation of the ROCK pathway, consistent with RNA-seq findings. Moreover, LA enhanced cellular proliferation, validating transcriptomic predictions. **Conclusions:** LA exerts a protective effect on intestinal epithelial integrity by stabilizing tight junctions, reducing MLC-2 phosphorylation, and promoting epithelial proliferation. These findings highlight a novel mechanism for LA and support its therapeutic potential in treating gastrointestinal disorders associated with “leaky gut” and mucosal injury.

## 1. Introduction

The intestinal mucosa is composed of single-layered columnar epithelia that form the body’s largest interface between the body interior and the external environment in the form of luminal contents [1]. The intestinal epithelium maintains its integrity and function by coordinating processes like proliferation, differentiation, migration, and apoptosis [2]. Intestinal epithelial cells are connected at the most apical region of their lateral membranes by specialized structures known as the apical junctional complex, which includes tight junctions (TJs) and adherens junctions. This complex connects the cells and restricts the movement of luminal contents, such as toxins and microbiota, while permitting the passage of ions, small molecules, and fluids through the paracellular space [3,4]. Myosin light chain II (MLC-2) is a crucial regulatory protein that influences paracellular permeability when the intestinal barrier is compromised. Research has indicated that the phosphorylation of MLC-2 leads to the contraction of the perijunctional actomyosin ring, thereby enhancing paracellular permeability by modulating the apical junctional complex. The myosin light chain kinase (MLCK) and Rho-associated coiled-coil containing protein kinase (ROCK) pathways play a crucial role in regulating the phosphorylation of MLC-2, which is vital for controlling intestinal permeability during intestinal disorders [5]. Intestinal mucosal permeability is elevated in gastrointestinal disorders, including inflammatory bowel diseases, [6] celiac disease, [7] and others, and this pathogenesis is associated with profound tissue anoxia [8]. In our previous study, anoxia/reoxygenation (A/R) injury elevated the paracellular permeability via TJ destabilization [9]. The disruption of TJ proteins was induced by increased MLC phosphorylation [9].

Larazotide acetate (LA) is a synthetic, eight amino acid peptide (H-Gly-Gly-Val-Leu-Val-Gln-Pro-Gly-OH) that is known to act as a TJ regulator capable of closing “leaky” interepithelial junctions [10,11,12]. Several clinical trials have confirmed the safety of this agent and suggest a potential beneficial effect of LA on gastrointestinal symptoms of patients with celiac disease. The epithelial monolayers pre-treated with LA showed protected paracellular permeability measured by transepithelial electrical resistance (TEER) and Lucifer Yellow flux assays in various barrier dysfunction models [11,12]. Additionally, the LA promoted actin rearrangement and redistribution of TJ proteins, including ZO-1, occludin, and claudins [11,12]. LA is developed and recognized as a zonulin antagonist that promotes the tightening of intestinal TJs [11,12]. Nevertheless, some debate remains regarding LA’s mechanism of action, including uncertainties surrounding zonulin’s identity, insufficient direct evidence, questions about LA’s specificity, and the potential for alternative cellular functions [13,14].

In this study, we investigated the mechanistic and functional role of LA in regulating epithelial integrity during cellular injury. Our findings indicate that apically administrated LA on A/R inured Caco-2BBe1 monolayer and leaky IPEC-J2 cells protected epithelial barrier functions via regulating TJ proteins and actin stabilization. Transcriptomic studies and subsequent validation assays suggest that LA regulates ROCK-mediated MLC-2 phosphorylation and enhances proliferation pathways, such as Wnt and Notch signaling. This study presents novel functions and cellular mechanisms of LA in the intestinal epithelium.

## 2. Materials and Methods

### 2.1. Cell Culture and Reagents

A human intestine Caco-2BBe1 (C2BBe1) cell line derived from Caco-2 cells was obtained from ATCC (Manassas, VA, USA) and grown in standard DMEM with 10% FBS and 0.01 mg/mL human transferrin (Invitrogen, Carlsbad, CA, USA) [15]. Intestinal porcine epithelial cell-jejunum 2 (IPEC-J2) cell lines were obtained from Dr. Anthony Blikslager at North Carolina State University. These cells were grown in standard DMEM media with 5% FBS, 1% insulin/selenium/transferrin, 0.5% epidermal growth factor, and 0.125% penicillin/streptomycin. The cells were maintained in a humidified 37 °C, 5% CO_2_ incubator.

Larazotide acetate (also known as AT1001) [16] was obtained from Innovative Biopharmaceuticals, Inc. (Raleigh, NC, USA). The final product was isolated as an acetate salt in a lyophilized form (>99% purity by HPLC/MS). The MLCK inhibitor peptide 18 and ROCK inhibitor Fasudil were purchased from Tocris Bioscience, Inc. (Bristol, UK) and STEMCELL Technologies (Vancouver, BC, Canada), respectively.

### 2.2. Anoxia/Reoxygenation Injury and Treatment of Reagents

To prepare cellular monolayers, C2BBe1 cells were plated at 100,000 cells per 24-well 0.4 µm pore-sized permeable supports (Corning, Tewksbury, MA, USA). Experiments were conducted 10–14 days post-plating. Larazotide acetate (0.01, 1, and 10 mM), peptide 18 (400 μM), or fasudil (100 μM) were pre-treated apically on the C2BBe1 monolayer before A/R injury.

To generate A/R injury, Caco-2BBe1 cells were placed in a modular incubator chamber (Billups-Rothenberg Inc., San Diego, CA, USA) and flushed with 95% N_2_/5% CO_2_ for 5 min. The modular chamber was then sealed airtight and placed in an incubator for 2 h at 37 °C. After 2 h, the cells were removed from the modular incubator chamber and placed in a 21% O_2_ normal environment. The anoxic injury was confirmed by detecting increased hypoxia-inducible factor-1α using a Western blot assay [9]. The cells were then incubated under normal environmental conditions for up to 4 h.

### 2.3. Leaky IPEC-J2 Model

For the leaky IPEC-J2 cells model, the media was composed of components enriched in the crypt, including 50% L-WRN (Wnt-3a, R-Spondin, Noggin) with advanced DMEM, 2 mM Glutamax, 10 mM HEPES (Thermo Fisher Scientific Inc., Waltham, MA, USA), 50 μg/mL Primocin (InvivoGen Corp., San Diego, CA, USA), 1 mM N-Acetyl-L-cysteine (MP Biomedicals., Irvine, CA, USA), and 250 μg/mL EGF (Thermo Fisher Scientific Inc., Waltham, MA, USA). The cells were passaged at least 5 times, considered a leaky IPEC-J2 model. The cells were plated at 100,000 cells per 24-well 0.4 µm pore-sized permeable supports (Corning, Tewksbury, MA, USA).

### 2.4. Measurement of Transepithelial Electrical Resistance

Cells on permeable support were used in this experiment once a TEER value of greater than 400 Ω cm^2^ for C2BBe1 or 30 Ω cm^2^ for IPEC-J2 cells was achieved. TEER was determined by a Chopstick Electrode Set (WPI, Sarasota, FL, USA) positioned on the apical and basal sides of the monolayers and attached to an Epithelial Volt Ohm Meter 2 (WPI, Sarasota, FL, USA). For all TEER measurements, the inserts were plated at an equal density; the readings were taken in triplicate per monolayer and averaged. The TEER recorded on blank inserts was subtracted from the TEER of inserts with cells.

### 2.5. Gel Electrophoresis and Western Blotting

The cell monolayers were washed with cold PBS and scraped in M-PER mammalian protein extraction reagent, including a protease inhibitor cocktail, 1 mM NaF, and 1 mM NaVO4 (Thermo Scientific, Rockford, IL, USA). Mem-PER^®^ Eukaryotic Membrane Protein Extraction Reagent Kit (Thermo Scientific, Rockford, IL, USA) was used to fractionate the membrane and cytosolic compartments according to the manufacturer’s protocol. Briefly, cells were sequentially treated with the provided permeabilization and solubilization buffers, with centrifugation steps used to separate the soluble cytosolic fraction from the solubilized membrane fraction. Protein analysis of extract aliquots was performed using a BCA protein assay kit (Thermo Scientific, Rockford, IL, USA). Tissue extracts, with amounts equalized by protein concentration, were mixed with XT sample buffer and a reducing agent (both Bio-Rad Laboratories Inc., Hercules, CA, USA) and boiled for 5 min. Lysates were loaded on a 4–10% SDS polyacrylamide gel, and electrophoresis was carried out according to standard protocols. Proteins were transferred to a PVDF membrane (Bio-Rad Laboratories Inc., Hercules, CA, USA) by using an electroblotting transfer apparatus. Membranes were blocked in Tris-buffered saline plus 0.1% Tween20 (TBST) with 5% BSA or skim milk and then incubated overnight with primary antibodies at 4 °C. The primary antibodies used were occludin, ZO-1 (Invitrogen, Carlsbad, CA, USA), pMLC-2, and MLC-2 (Cell signaling, Danvers, MA, USA). After washing in TBST, membranes were incubated with horseradish peroxidase conjugated secondary antibodies and developed for protein visualization with chemiluminescence (Thermo Scientific, Rockford, IL, USA).

### 2.6. Immunofluorescence Microscopy Analysis

The cells grown on permeable supports were fixed with cold (−20 °C) absolute methanol and stored at −80 °C until used. The cells were thawed, rinsed in PBS, blocked with protein block serum-free (DakoCytomation, Via Real Carpinteria, CA, USA), and incubated overnight at 4 °C with primary antibodies in antibody diluent (DakoCytomation, Via Real Carpinteria, CA, USA). The cells were washed thoroughly with PBS and incubated in secondary antibodies conjugated with fluorescent dyes, and nuclei were stained with DAPI (Thermo Scientific, Rockford, IL, USA). Following rinsing in PBS, the cells were mounted on fluorescent mounting media (DakoCytomation, Via Real Carpinteria, CA, USA) and examined with an Olympus IX83 Inverted Motorized Microscope with cellSens software (Olympus corporation, Version 4.2.1, Shinjuku, Tokyo, Japan).

### 2.7. Total RNA Isolation and RNA Sequencing

The confluent C2BBe1 monolayers were subjected to be administrated with or without 10 mM LA before undergoing A/R injury. After 2 h of A/R injury, as previously described, we collected the cell lysate at either 1 or 22 h of reoxygenation. Control cells, both treated and untreated with 10 mM LA, were collected at 3 or 24 h. The total RNAs were isolated using RNeasy Plus Mini Kit (Qiagen, Hilden, Germany) and submitted for next-generation RNA sequencing performed by Novogenes, Inc. (Sacramento, CA, USA) (*n* = 3/group). Briefly, the process followed mRNA enrichment and purification, fragmentation and reverse transcription, adaptor litigation, and PCR amplification to prepare the cDNA library. The library preparations were sequenced on an Illumina Hiseq 2500 platform. The QC analysis (Appendix A) and differential expression analysis of each group were performed using the DESeq2 R package (2_1.6.3). The resulting *p*-values were adjusted using Benjamini and Hochberg’s approach for controlling the False Discovery Rate (FDR). Genes with an adjusted *p*-value < 0.05 found by DESeq2 were assigned as differentially expressed.

### 2.8. GO, KEGG, and GSEA Enrichment Analysis of Differentially Expressed Genes

Gene Ontology (GO) enrichment analysis of differentially expressed genes was implemented by the clusterProfiler R package, in which gene length bias was corrected. GO terms with corrected *p*-value less than 0.05 were considered significantly enriched by differentially expressed genes.

The Kyoto Encyclopedia of Genes and Genomes (KEGG) is a database resource for understanding the high-level functions and utilities of biological systems, such as cells, organisms, and ecosystems, from molecular-level information, particularly large-scale molecular datasets generated by genome sequencing and other high-throughput experimental technologies (http://www.genome.jp/kegg/, URL accessed on 8 July 2024). We used clusterProfiler R package to test the statistical enrichment of differential expression genes in KEGG pathways. GSEA was performed with Dr. Tom’s online platform (https://biosys.innomics.com, URL accessed on 15 September 2024).

### 2.9. Proliferation and Migration Assays

The proliferation of non-treated or LA-treated (10 mM) C2BBe1 cells was evaluated by the CCK-8 assay kit (Dojindo Molecular Technologies, Rockville, MD, USA) according to the manufacturer’s instructions. 10 mM LA was administered to cells seeded in 96-well culture plates (3000 cells/well) every other day. Viable cells were evaluated with the CCK-8 Assay Kit for five consecutive days.

Migration was assessed by seeding C2BBe1 cells in 12-well culture-inserts (100,000 cells/well) (ibidi GmbH, Gräfelfing, Germany). Cells were grown to 100% confluence before the inserts were removed. For the serum-free condition, 16 h before removing the insert, the cells will be cultured with serum-free media to prevent cell proliferation. After removing the inserts, the 10 mM LA was added to the serum-free medium to assess migration. Images were captured at 0, 4, 8, 24, and 48 h with an Axio Vert A1 microscope (Carl Zeiss AG, Oberkochen, Germany), and the closing of the wound was analyzed by measuring migration distance with ImageJ software (Version 1.54i, https://imagej.net/ij. URL accessed on 11 September 2024).

### 2.10. Statistical Analysis

Data are reported as mean ± standard error. Differences between groups were tested with two- or one-way ANOVA with Tukey or LSD post hoc tests (GraphPad Prism 10). Where appropriate, the difference between the two groups was assessed with a T-test (GraphPad Prism 10). A *p*-value of <0.05 was considered significant for all statistical analyses.

## 3. Results

### 3.1. The TJ Barrier Is Tightened with the Pretreatment of LA in Leaky Intestinal Cell Models

The LA is known as a TJ regulator that improves intestinal barrier functions [15]. To study the role of LA in epithelial permeability during A/R injury, Caco-2BBe1 monolayers were apically administrated with LA (0.1, 1, and 10 mM) and exposed to anoxia (95% N_2_/5% CO_2_) for 2 h, followed by reoxygenation for up to 4 h (Figure 1A). During the A/R injury model, the permeability of cell monolayers was monitored following A/R injury by using TEER measurement (Figure 1B). Anoxic injury resulted in diminished TEER values compared with control cells (*p* < 0.05), which is indicative of impaired barrier function. During the reoxygenation, the TEER recovered within 4 h. The TEER value on the monolayer pretreated with 0.1, 1, and 10 mM LA significantly protected the epithelial barrier during anoxic injury. Remarkably, the epithelial monolayer exposed to 10 mM LA exhibited a greater increase than the control after 4 h of reoxygenation. (*p* < 0.01). Based on our dose–response findings, a 10 mM concentration of LA was used to achieve a maximal protective effect in our in vitro system, although we recognize this concentration may not directly translate to in vivo clinical dosage. Thus, we used 10 mM of LA for the last of the experiments.

Next, we intend to assess the effectiveness of LA on the non-immortalized, non-cancerous primary-derived IPEC-J2 cell line from the piglet’s jejunum. For this study, we created a leaky gut model using IPEC-J2 cells, which involved culturing them for over five passages in enteroid culture media to mimic a crypt-like environment. This leaky IPEC-J2 culture model maintains significantly lower and physiologically relevant TEER (30–80 Ω·cm^2^) compared to the higher TEER level (4000–5000 Ω·cm^2^) in the conventional IPEC-J2 culture (Appendix A). In this model, treatment with 10 mM of LA notably improved the TEER after 8 h (*p* < 0.01). Based on these data, we are able to confirm the role of LA in providing barrier protection during in vitro A/R injury or the IPEC-J2 leaky gut model.

### 3.2. TJ Proteins Are Protected by the Pretreatment of Larazotide Acetate During A/R Injury

We have previously shown that the TJ proteins occludin and ZO-1 were internalized after anoxic injury and recovered during reoxygenation [9]. In this study, we aimed to investigate the role of LA in regulating TJ proteins by utilizing Western blotting of fractionated membrane and cytosol extract and performing immunofluorescence microscopy one-hour following anoxic injury. In control cells, transmembrane TJ protein occludin was localized in the membrane fraction, and cytosol plaque protein ZO-1 was predominantly expressed in the cytosolic fraction (Figure 2A). While the expression of ZO-1 in the cytosolic fraction was not changed during A/R injury, the expression of occludin in the membrane fraction was significantly (*p* < 0.05) reduced after anoxic injury, and pretreatment with 10 mM LA was protected with the reduction in its expression in the membrane fractions (Figure 2A).

To further assess the redistribution of TJ proteins by LA during A/R injury, Caco-2BBe1 monolayers were evaluated by immunofluorescence microscopic analysis (Figure 2B). As shown in control cells, the TJ proteins occludin and ZO-1 were distributed to the region of the interepithelial TJs, outlining cell–cell contact edges. However, A/R injury induced pronounced reorganization of ZO-1 with a “wavy line” pattern (arrows), which is indicative of disruption of TJ integrity. In addition, occludin was dramatically internalized into cytoplasmic vesicles (arrowheads) 1 h after anoxic injury. However, the monolayers pre-treated with 10 mM LA were protected from disruption of TJ during A/R injury (Figure 2B). Since the administration with LA also affected the reorganization of cytoskeleton protein F-actin, which is predominantly associated with TJ stability [11,12], we also studied the role of LA in F-actin during A/R injury. The F-actin was dramatically reorganized during the A/R injury compared to the control monolayer. The monolayer pretreated with 10 mM LA was prevented from this reorganization of F-actin. Taken together, TJ stability was protected by pretreatment with LA in Caco-2BBe1 cells during A/R injury.

### 3.3. Transcriptomic Analyses Revealed Differently Expressed Genes in the Cells Pretreated with LA

Although the role of LA in preserving intestinal mucosal integrity has been widely studied, its mechanisms and potentially varied functions remain incompletely understood. To investigate these aspects, we performed next-generation RNA sequencing analysis to assess the possible mechanisms and cellular functions of LA under both normal conditions and A/R injury. Cells were pretreated with 10 mM of LA, both in control (CT) and A/R injured groups, prior to injury, and were collected 3 and 24 h post-prement and 1 h and 22 h after 2 h A/R injury (2I/1R and 2I/22h), respectively. In the LA-treated C2BBe1 cells, significant transcriptomic alterations were observed compared to untreated CT or A/R injured cells after 3 h (CT_LA_3 h vs. CT_3 h; A/R + LA_2I/1R vs. A/R_2I/1R), as demonstrated by Principal component analysis (PCA) and cluster analysis of differentially expressed genes (DEGs). However, after long-term incubation with LA (24 h), LA did not produce significant transcriptomic differences in either CT or A/R injury (CT_LA_24 h vs. CT_24 h; A/R + LA_2I/22R vs. A/R_2I/22R) (Figure 3A,B), suggesting that the effects of LA were not maintained over time. The analysis of DEGs, GO enrichment, and KEGG pathways analysis comparing CT_LA_3 h with CT_3 h yielded results similar to those from A/R + LA_2I/1R versus A/R_2I/1R. Therefore, we present the analytical data for A/R + LA_2I/1R compared to A/R_2I/1R here.

Out of the 338,714 annotated genes, we identified a total of 3053 statistically significant DEGs encoding mRNA (1477 upregulated and 1576 downregulated) (Appendix A, q-value ≤ 0.05 − adjusted *p*-value that accounts for the false discovery rate) between A/R + LA_2I/1R and A/R_2I/1R. We selected 16,245 genes by expression level (TPM > 0) to plot the Venn diagram, identifying 531 genes uniquely expressed in A/R + LA_2I/1R and 530 genes uniquely expressed in A/R_2I/1R, while 15,184 genes were expressed in both groups (Figure 3C). Next, we plotted the Volcano map with DEGs, identifying the top 20 upregulated (red dots) and 24 downregulated (green dots) genes (log2FC ≥ 1, q-value ≤ 0.05) (Figure 3D, Appendix A). The upregulated genes (e.g., AMOTL2, DDIT4, EGR1, etc.) were related to epithelial barrier function, proliferation, and differentiation, while the downregulated genes (e.g., ITGA2, MMP1, CEMIP, etc.) were associated with cell migration and focal adhesion.

### 3.4. GO Enrichment Analyses Revealed Changes in Cellular Processes Resulting from LA Pretreatment on the A/R Injured Intestine Epithelium

We utilized GO enrichment to investigate the functional terms of the enriched gene sets in A/R + LA_2I/1R versus A/R_2I/1R (Figure 4, Appendix A). Within the GO biological process, the top 10 significant terms featuring upregulated genes include categories related to small GTPase (such as small GTPase-mediated signal transduction and its regulation), cell polarity (including establishment and maintenance of cell polarity), and protein phosphorylation (e.g., protein autophosphorylation) (Figure 4A). In the top 10 GO molecular functions, we identified terms associated with protein phosphorylation (like protein serine/threonine kinase activity), small GTPase (including Ras GTPase and Rho GTPase binding), and transcriptional activity (such as transcription corepressor and transcription factor activities) (Figure 4B). The GO cellular components analysis highlights terms connected to intercellular junctions (like cell–cell junctions and the apical junction complex) and migration (including cell leading edge and lamellipodium) (Figure 4C). Conversely, the GO terms with downregulated genes were primarily linked to transcriptional and translational processes (like ribonucleoprotein complex biogenesis and RNA splicing) (Figure 4D). Thus, the GO term analysis indicated that LA supports the epithelial barrier by enhancing small GTPase and protein phosphorylation activities at the intercellular junctions. Additionally, the emphasis on migration-related GO terms suggests it may also influence cellular migration activities.

### 3.5. KEGG Pathway and GSEA Assay Showed the Alteration of Critical Cellular Pathways by LA Pewtreatment in A/R Injured Intestinal Epithelial Cells

The KEGG pathway analysis was performed to explore cellular mechanism pathways that were altered by LA on the A/R injured C2BBe1 cells. Interestingly, pathways related to proliferation, such as the Wnt signaling pathway, Notch signaling pathway, and cancer pathways, were underscored by upregulated genes (Figure 5A). Additionally, the intestinal barrier-associated pathways, such as TJs and adherens junctions, were also revealed (Figure 5A). To strengthen our findings, we additionally conducted GSEA using Hallmark gene sets (Figure 5B). We observed that the gene sets related to Wnt and Notch signaling, as well as those involving epithelial barrier and polarity (such as apical_surface and apical_junctions), were elevated in the LA-pretreated A/R injured cells compared to the A/R injured monolayer (Figure 5C). These results suggest that LA may also regulate cellular proliferation while enhancing the functions of apical junctional barriers.

### 3.6. Pretreament with LA Reduced Anoxia-Induced Phosphorylation of MLC-2 During A/R Injury via ROCK Pathway

Based on our GO enrichment results, LA regulated the apical junctional complex via small GTPase and protein phosphorylation (Figure 4). It is a well-known mechanism that the disruption of the TJ barrier occurs due to increased phosphorylation of MLC-2, as demonstrated in our prior findings regarding A/R injury [9]. Therefore, we next decided to evaluate the mechanistic role of LA in regulating the phosphorylation of MLC-2, which is crucial for changes in the epithelial barrier during A/R injury. The ratio of phosphorylated MLC-2 was significantly (*p* < 0.05) elevated at 1 h after anoxic injury, and pretreatment with 10 mM of LA significantly (*p* < 0.01) reduced this phosphorylation (Figure 6A,B). Thus, these results indicated that pretreatment with LA protects the TJ barrier by reducing the phosphorylation of MLC-2 during A/R injury.

MLCK and ROCK regulate the phosphorylation of MLC-2 to regulate barrier integrity in response to impairment of the intestinal epithelium [5]. Additionally, ROCK also inhibits the dephosphorylation of MLC by inactivating the MLC phosphatase (Figure 6C) [5]. Peptide 18 and fasudil were utilized to inhibit the enzymic activity of MLCK and ROCK, respectively. Inhibition of MLCK alone with peptide 18 increased TEER compared to A/R injured cells. The combined peptide 18 and LA during injury exhibited a synergistic effect, as the cumulative effect resulted in a greater enhancement of TEER than the sum of the individual effect. Epithelial resistance was also increased in ROCK inhibition with fasudil pretreatment. However, the combined administration of larazotide and fasudil did not show a synergistic or additive increase in resistance (Figure 6D). This indicated that the LA and fasudil shared their mechanism and did not produce additive or synergistic effects on TEER. Our findings suggest that the reduced phosphorylation of MLC by larazotide is likely regulated through ROCK activity. It was further supported by RNAseq analysis, which highlighted ROCK regulators, specifically small GTPases, in the GO enrichment assay (Figure 4).

### 3.7. Treatment of LA in C2BBe1 Cells Enhanced Cellular Proliferation and Migration

Given that the RNAseq analysis indicated a potential role for LA in cellular proliferation and migration, we examined LA’s effects on the proliferation and migration of C2BBe1 cells. The treatment with 10 mM LA showed a significantly increased proliferation (*p* < 0.01) compared to the non-treated C2BBe1 cells using the CCK-8 analysis (Figure 7A). However, the 10 mM LA did not demonstrate significant migration effects on pre-starved (serum-free) C2BBe1 monolayers, although it significantly increased wound closure in a 10% FBS containing media, which cannot exclude the effects from proliferation (Figure 7B). Consequently, the LA not only maintains the intercellular junctional barrier but also promotes cellular proliferation processes, as validated by the RNA sequencing assay data and subsequent proliferation analysis, suggesting the mucosal healing effects of LA.

## 4. Discussion

In this study, administration of LA in A/R-injured Caco-2BBe1 or leaky IPEC-J2 cell models significantly elevated TEER compared to non-treated cells. Additionally, the TJ protein actin structure was intensely disrupted during A/R injury of C2BBe1 cells, and the monolayer pretreated with LA was protected from these disruptions. The RNAseq analysis and cell functional validation studies also uncovered novel potential mechanistic pathways and cellular functions through which LA regulates the TJ barrier and proliferation.

The mechanism of action of LA remains under study. It demonstrates a protective effect against the increased permeability induced by AT-1002, a hexamer of zonula occludens toxin (Zot) derived from Vibrio cholerae, or zonulin, the human counterpart of Zot [11,16,17]. LA was initially identified as an octapeptide matching the N-terminal sequence of zonulin, which is the human equivalent of Zot [11]. Thus, it was proposed as a zonulin antagonist that stabilizes TJs and maintains intestinal barrier integrity by inhibiting the zonulin binding to its receptor. However, evidence suggests LA derives from an immunoglobulin sequence, questioning its relevance to zonulin biology [14]. Moreover, the belief that zonulin is a reliable biomarker of intestinal permeability is challenged by findings that ELISA kits do not accurately measure zonulin (pre-haptoglobin 2) but detect unrelated proteins like properdin [18]. Despite these challenges, Larazotide has demonstrated barrier-protective and therapeutic effects in various in vitro and in vivo studies [11,12,15,19]. In terms of safety, LA has been extensively studied in Phase 1, 2, and 3 clinical trials for celiac disease. Across these studies, the compound has demonstrated a favorable safety profile, showing no significant difference in adverse events compared to placebo and no evidence of systemic toxicity, reinforcing its potential as a locally acting, non-systemic therapeutic agent [20,21]. Therefore, understanding the mechanism of action of Larazotide is crucial for future approaches.

Our study highlights a critical mechanism through which LA preserves intestinal barrier integrity by modulating the phosphorylation status of MLC-2, a key regulator of TJ dynamics. A/R injury significantly increased MLC-2 phosphorylation, contributing to cytoskeletal contraction and destabilization of TJ proteins such as occludin and ZO-1—a finding consistent with prior studies on MLC-2–mediated epithelial barrier dysfunction [9]. Pretreatment with LA markedly reduced this phosphorylation, thereby mitigating disruption of the TJs (Figure 7). This protective effect appears to be mediated via the ROCK pathway, as inhibition of ROCK with fasudil mirrored the effects of LA and abolished any synergistic response when combined (Figure 7), suggesting overlapping mechanisms. Conversely, combined inhibition of MLCK and LA exhibited a synergistic effect on barrier restoration, implying that MLCK and LA act through distinct but complementary signaling cascades. Taken together, these findings suggest that LA exerts its barrier-stabilizing effect by attenuating MLC-2 phosphorylation through modulation of the ROCK pathway. Additionally, it sheds light on possible combination treatment options between LA and MLCK inhibitors for treating gastrointestinal disorders.

The RNA sequencing data further supported this mechanistic model, showing enrichment in genes associated with small GTPase signaling and cytoskeletal regulation following LA administration (Figure 5). Specifically, enriched GO terms included Rho GTPase binding, Ras signaling, and serine/threonine kinase activity—molecular functions closely associated with cytoskeletal regulation and TJ remodeling. These findings reinforce the hypothesis that LA exerts its barrier-protective effects by modulating upstream regulators of ROCK, such as Rho and Ras family GTPases. The enrichment of apical junction and cell–cell adhesion genes with upregulated DEGs, alongside decreased expression of genes associated with focal adhesion disassembly and extracellular matrix remodeling, suggests a global reinforcement of epithelial barrier structure. Taken together, our findings propose that LA promotes epithelial stability through both post-translational (small GTPase- or kinase-mediated) and transcriptional regulation of the cytoskeletal and junctional networks.

Importantly, transcriptomic data also pointed to the activation of epithelial proliferation pathways. KEGG and GSEA pathway analyses revealed significant upregulation of Wnt and Notch signaling pathways—both of which are central to epithelial renewal and stem cell maintenance. These findings were functionally validated by in vitro proliferation assays, which demonstrated a significant increase in epithelial cell growth following LA treatment (Figure 7A). The upregulated expression of genes such as EGR1, DDIT4, and AMOTL2, known regulators of cell proliferation and differentiation, supports LA’s dual role in both maintaining junctional integrity and enhancing epithelial restitution. This proliferative effect may be particularly advantageous in pathological contexts where epithelial turnover is impaired, further underscoring LA’s therapeutic potential. This would make LA a promising therapeutic option for mucosal healing in mucosal inflammatory disorders, such as inflammatory bowel disease and necrotizing enterocolitis. However, it is important to consider the potential long-term implications of stimulating this pathway. Chronic intestinal inflammation or repeated injury cycles, which demand sustained proliferative repair, have been linked to an increased risk of dysplasia and tumorigenesis. Therefore, while LA’s pro-proliferative effects are advantageous for acute barrier repair, further investigation is warranted to understand its long-term safety profile in chronic conditions that involve persistent mucosal injury and repair.

Furthermore, the A/R model used in our study is highly relevant to a number of critical human gastrointestinal disorders characterized by ischemia–reperfusion (I/R) injury. Clinical conditions such as NEC in premature infants, acute mesenteric ischemia resulting from arterial or venous occlusion, and complications from major surgical procedures like intestinal transplantation and aortic aneurysm repair all share I/R injury as a core pathophysiological mechanism. In these settings, the temporary loss of blood flow (ischemia) followed by its restoration (reperfusion) triggers a potent inflammatory cascade and the production of reactive oxygen species, leading to severe damage to the intestinal mucosal barrier. This breach in the barrier allows for the translocation of luminal bacteria and toxins into the bloodstream, which can result in systemic inflammatory response syndrome, sepsis, and multi-organ failure [22,23]. Our findings suggest that LA could be a valuable therapeutic agent in these clinical scenarios by stabilizing tight junctions and preserving barrier integrity during the critical reperfusion phase, thereby mitigating the severe downstream consequences of I/R injury.

While our study provides valuable insights into the protective mechanisms of Larazotide Acetate on intestinal enterocytes, we must acknowledge the limitations of our in vitro Caco-2 and IPEC-J2 model. These monoculture systems, though robust, do not fully recapitulate the complex cellular and structural architecture of the native intestinal barrier, which includes a diverse array of cell types that contribute to the mucosal defense and repair processes. Therefore, future work is essential to validate these findings in more physiologically sophisticated systems. The logical progression would be to utilize 3D intestinal organoids to study these intercellular interactions, followed by in vivo studies in animal models of ischemia–reperfusion injury. Another limitation is our prophylactic design. We showed that pre-treatment with Larazotide Acetate protects the barrier from injury, but not whether it can therapeutically repair existing damage. Future studies should assess post-injury application to determine its full clinical utility. Such research will be critical for confirming the therapeutic efficacy of LA and advancing its potential translation into clinical applications for acute intestinal injuries.

## 5. Conclusions

This study demonstrates that LA effectively protects intestinal epithelial barrier integrity during A/R injury by stabilizing TJ proteins and reducing MLC-2 phosphorylation, most likely through regulation of the ROCK pathway. Transcriptomic analysis further revealed that LA activates key signaling pathways involved in cytoskeletal organization, TJ maintenance, and epithelial proliferation. These combined effects highlight LA’s dual role in reinforcing barrier structure and promoting mucosal repair. Overall, our findings provide novel mechanistic insights supporting LA as a promising therapeutic agent for gastrointestinal diseases characterized by increased intestinal permeability and mucosal injury.

## Figures and Tables

**Figure 1 biomedicines-13-02483-f001:**
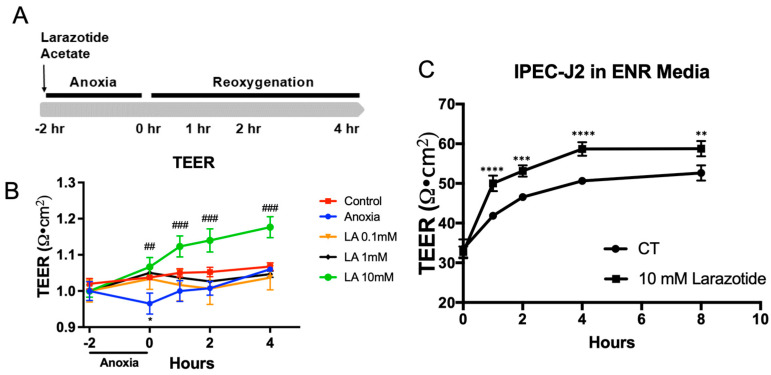
Effects of larazotide acetate (LA) on paracellular permeability in A/R injury C2BBe1 or IPEC-J2 leaky gut model. (**A**) Schematic drawing showing the schedule of LA pretreatment and TEER measurement in A/R injury of C2BBe1 cells. (**B**) Pretreatment with 10 mM larazotide significantly increases the TEER compared to the non-treated A/R injured C2BBe1 monolayer. *, *p* < 0.05, control vs. anoxia; ^##^, *p* < 0.01 and ^###^, *p* < 0.001 anoxia vs. LA 10 mM. (**C**) The leaky IPEC-J2 monolayer treated with 1 mM LA had a significantly higher TEER compared to the non-treated one. *n* = 4–6 wells/group, **, *p* < 0.01, ***, *p* < 0.001, ****, *p* < 0.0001 control (CT) vs. 10 mM LA.

**Figure 2 biomedicines-13-02483-f002:**
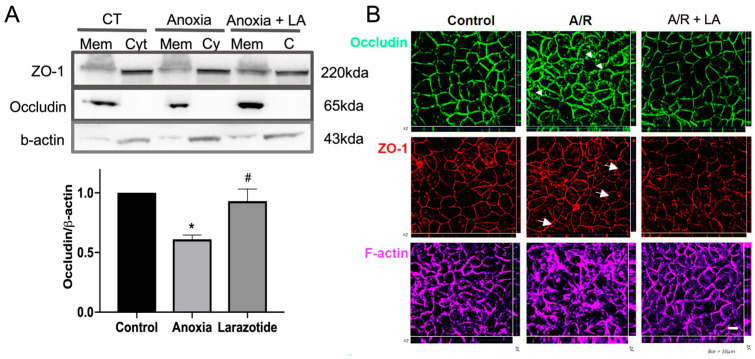
The localization of tight junction (TJ) proteins was analyzed by immunoblotting and IF analyses after 1 h of reoxygenation, following anoxic injury on C2BBe1 cells with or without LA. (**A**) The transmembrane protein occludin was significantly (*p* < 0.05) reduced in anoxic injured cells compared to control cells in the membrane fraction. However, the occludin was significantly (*p* < 0.05) increased by 10 mM of LA compared to non-treated anoxic injured cells. *, *p* < 0.05, control vs. anoxia; ^#^, *p* < 0.05, anoxia vs. LA 10 mM. (**B**) Permeable support membranes were fixed and stained for ZO-1 (red), occludin (green), and F-actin (purple). Immunolocalization of ZO-1 and occludin was analyzed by Z stack 3D analysis. Whereas non-treated anoxic injured cells showed disruption of occludin (arrowheads), ZO-1 (arrows), and F-actin, LA-pretreated cells revealed well-organized TJ proteins and F-actin.

**Figure 3 biomedicines-13-02483-f003:**
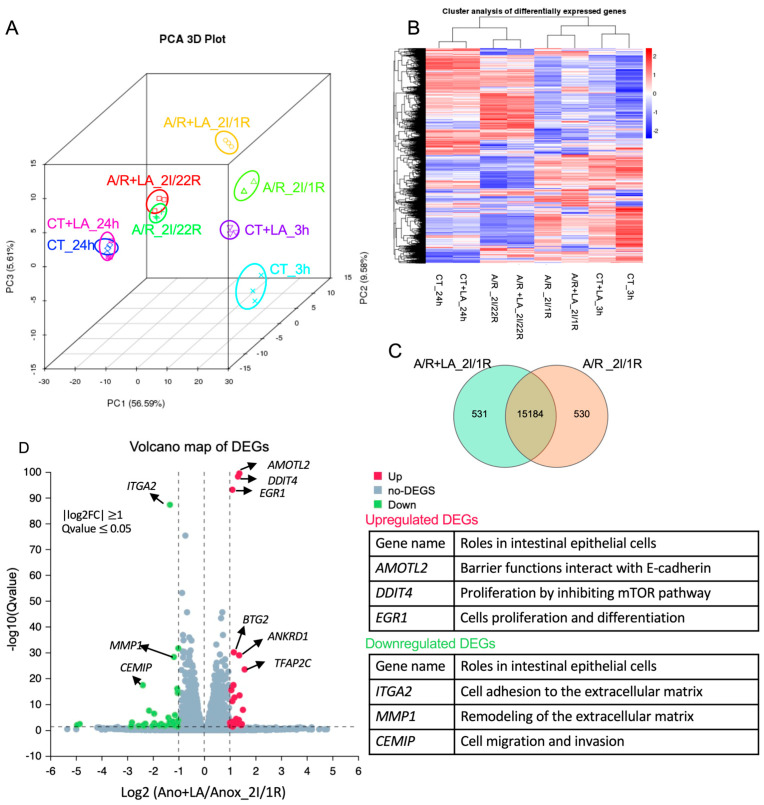
Differential expression analyses using the next-generation RNA-seq. (**A**,**B**). Principal component analysis (PCA) and Hierarchical clustering of differentially expressed genes (DEGs). (**C**) Number of common and unique DEGs in A/R injured C2BBe1 cells pretreated with LA compared to the A/R injured ones without LA using the Venn diagram. (**D**) Volcano plot showing DEGs in A/R injured C2BBe1 cells pretreated with LA versus the A/R injured ones without LA. The most significantly upregulated or downregulated genes are labeled with their names, and their roles in intestinal epithelial cells are summarized in the tables.

**Figure 4 biomedicines-13-02483-f004:**
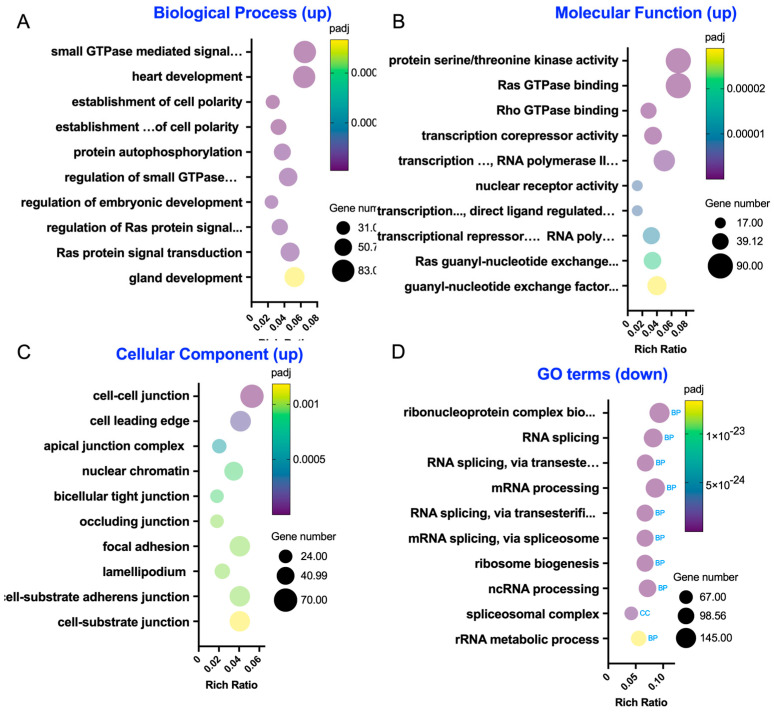
Gene Ontology (GO) enrichment analysis in the A/R injured C2BBe1 cells pre-treated with LA compared to the A/R injured ones without LA. (**A**–**C**) The 10 most significantly enriched GO terms in biological process, molecular function, and cellular component with upregulated differentially expressed genes (DEGs). (**D**) The 10 most significantly enriched GO terms with downregulated DEGs.

**Figure 5 biomedicines-13-02483-f005:**
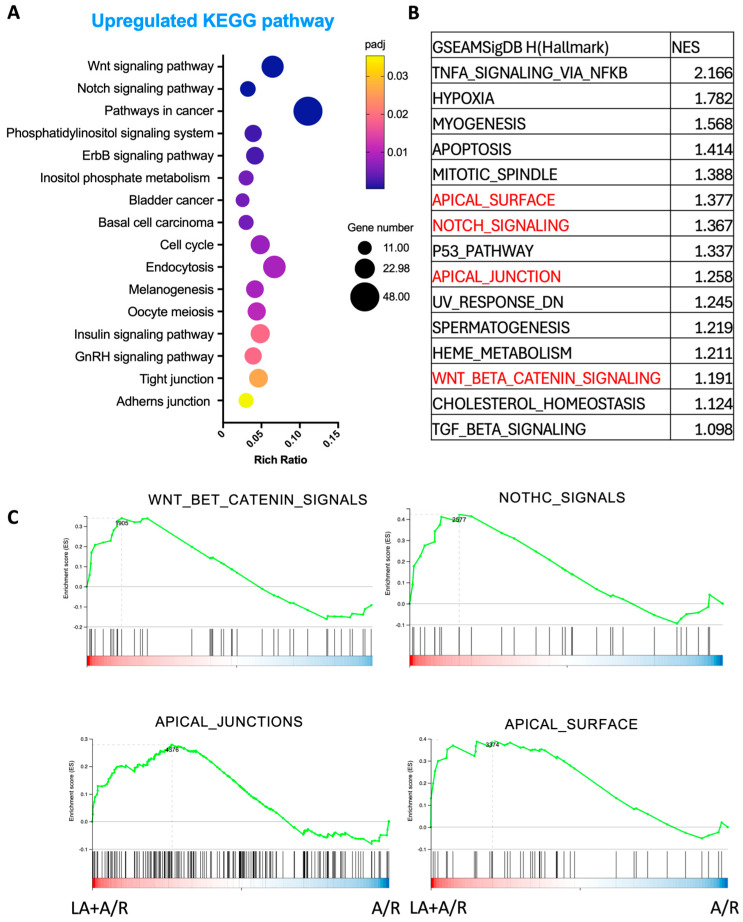
Kyoto Encyclopedia of Genes and Genomes (KEGG) pathway and Gene set enrichment analysis (GSEA) analysis in the A/R injured C2BBe1 cells pretreated with LA compared to the A/R injured ones without LA. (**A**) The 16 most significantly enriched KEGG pathways with upregulated DEGs. (**B**) The 15 highest-ranked gene sets were determined by normalized enrichment score (NES). (**C**) The selected gene sets for important signaling path-ways were highlighted with a GSEA plot.

**Figure 6 biomedicines-13-02483-f006:**
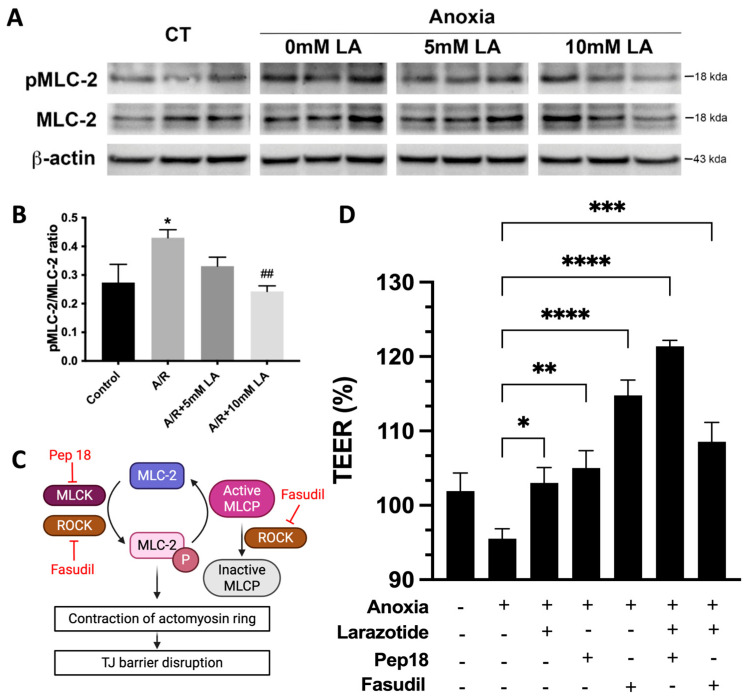
LA decreased the pMLC-2 levels increased by anoxia injury, likely through ROCK inhibition. (**A**,**B**). pMLC-2 levels in anoxically injured C2BBe1 were significantly decreased in LA-pretreated cells, as shown by immunoblot assay. *, *p* < 0.05, control vs. anoxia; ^##^, *p* < 0.01, anoxia vs. LA 10mM. (**C**) Schematic diagram illustrating how ROCK and MLCK regulate MLC-2 phosphorylation, along with their respective inhibitors. Created in BioRender. Jin, Y. (2025) https://BioRender.com/ptfl7qc. (accessed on 11 October 2011) (**D**) A pharmacological inhibition study was conducted using MLCK inhibitor (Pep 18) and ROCK inhibitor (Fasudil) in combination with LA. *, *p* < 0.05, **, *p* < 0.01, ***, *p* < 0.001, ****, *p* < 0.0001 vs. anoxia.

**Figure 7 biomedicines-13-02483-f007:**
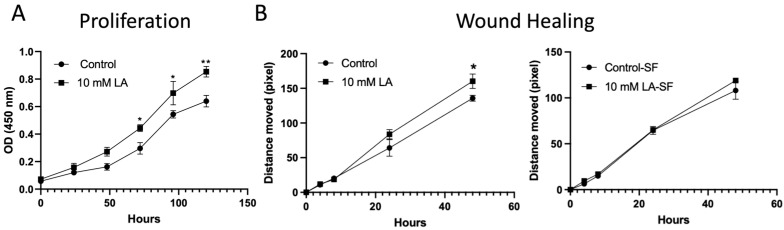
Proliferation and wound healing assays were conducted on C2BBe1 cells with or without 10 mM LA. (**A**) Cells treated with LA showed a significant increase in proliferation, as measured by the CCK-8 assay. (**B**) Pre-starved C2BBe1 cells treated with LA did not exhibit a significant increase in wound healing, whereas cells cultured in conventional media showed a notable improvement. *, *p* < 0.05, **, *p* < 0.01, vs. control.

## Data Availability

The raw sequencing data for the RNA-seq experiments presented in this article are openly available in the NCBI Sequence Read Archive (SRA) under the BioProject accession number PRJNA1311078.

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
