# Peer review of "Larazotide Acetate Protects the Intestinal Mucosal Barrier from Anoxia/Reoxygenation Injury via Various Cellular Mechanisms"

_biomedicines, 2025, doi:10.3390/biomedicines13102483_

Round 1
Reviewer 1 Report
Comments and Suggestions for Authors
The paper investigates how Larazotide acetate (LA) protects intestinal barriers during A/R injury using cell models and transcriptomic analysis. The work provides valuable mechanistic insights into LA but requires technical refinement and contextual depth.
1. Justify the high LA concentration (10 mM) used throughout the study, as this exceeds typical clinical doses. Provide rationale or dose-response data.
2. Figure 1B: Add statistical markers (e.g., asterisks) to explicitly show significance between LA-treated vs. untreated groups at 2h/4h reoxygenation.
3. Clarify the mismatch: Samples were collected at 3h/24h (L156-159), but A/R injury groups are labeled "21hR" (Fig 3A). Reconcile timepoints.
4. Specify the number of biological replicates for RNA-seq and include QC metrics (e.g., RIN values).
5. Correction: "tarazotide" → "Larazotide" (Fig 2 legend)
"C2Bbe1" → "C2BBe1" (L244)
"has significantly increase" → "had significantly increased" (L333)
"Recoxygenation" → "Reoxygenation"
6. Ensure all supplementary materials (Fig S1, Tables S1-S3) are included and cited correctly.
7. Fig 5B: Define "NES" in legend
8. Abbreviations: Add "PCA" (Principal Component Analysis) to list
9. Data Availability: Provide accession number for RNA-seq data in public repository
10. Discussion: (1) Discuss if increased proliferation is beneficial or potentially tumorigenic in barrier repair contexts. (2) Strengthen the clinical relevance of A/R injury to human GI disorders (e.g., ischemia-reperfusion).
Author Response
Dear Reviewer,
We sincerely thank you for the thoughtful and constructive comments on our manuscript. We have revised the paper accordingly, improving clarity, methodological detail, and clinical context. Below, we provide a detailed, point-by-point response to each reviewer’s comment. Changes have been highlighted in the revised manuscript.
- Comment: Justify the high LA concentration (10 mM) used throughout the study, as this exceeds typical clinical doses. Provide rationale or dose-response data.
Response: Thank you for this important question. We acknowledge that the 10 mM concentration is higher than typical clinical doses. Our dose-response data, presented in Figure 1B, show a clear, dose-dependent protective effect of LA on the intestinal barrier during anoxic injury, with the 10 mM concentration providing the most significant protection in our model. We have added a sentence to the manuscript to clarify that this concentration was chosen to elicit a maximal response in our in vitro system, which may not directly translate to in vivo clinical dosage.
- Comment: Figure 1B: Add statistical markers (e.g., asterisks) to explicitly show significance between LA-treated vs. untreated groups at 2h/4h reoxygenation.
Response: We appreciate this suggestion. This is about Figure 1C. We have revised Figure 1C to include asterisks indicating the statistical significance between the LA-treated and untreated groups.
- Comment: Clarify the mismatch: Samples were collected at 3h/24h (L156-159), but A/R injury groups are labeled "21hR" (Fig 3A). Reconcile timepoints.
Response: Thank you for pointing out this inconsistency. The "22hR" label refers to 22 hours of reoxygenation following a 2-hour anoxic injury that referred to 2I/22hR, resulting in a total experimental time of 24 hours, comparable to the 24-hour control group. We have clarified this in the methods and results section and figure legends to ensure consistency.
- Comment: Specify the number of biological replicates for RNA-seq and include QC metrics (e.g., RIN values).
Response: We have amended the methods section to include the number of biological replicates (n=3/group) used for the RNA-seq analysis. We have also added the QC metrics as a supplementary table (S1 table) to provide quality control metrics for the RNA samples.
- Comment: Correction: "tarazotide" → "Larazotide" (Fig 2 legend) "C2Bbe1" → "C2BBe1" (L244) "has significantly increase" → "had significantly increased" (L333) "Recoxygenation" → "Reoxygenation".
Response: We have corrected these typographical errors.
- Comment: Ensure all supplementary materials (Fig S1, Tables S1-S3) are included and cited correctly.
Response: We have double-checked all supplementary materials and their citations to ensure they are correctly included and referenced in the text.
- Comment: Fig 5B: Define "NES" in legend.
Response: We have defined "NES" as "Normalized Enrichment Score" in the caption for Figure 5B.
- Comment: Abbreviations: Add "PCA" (Principal Component Analysis) to list.
Response: We have added "PCA" to the list of abbreviations.
- Comment: Data Availability: Provide accession number for RNA-seq data in public repository.
Response: We have included the accession number for the RNA-seq data in the Data Availability Statement.
- Comment: Discussion: (1) Discuss if increased proliferation is beneficial or potentially tumorigenic in barrier repair contexts. (2) Strengthen the clinical relevance of A/R injury to human GI disorders (e.g., ischemia-reperfusion).
Response: We have expanded the discussion to address these points. We now discuss the dual nature of increased proliferation in barrier repair, acknowledging its benefits while also considering the potential long-term risks. We have also added a more detailed explanation of how the A/R injury model mimics the ischemia-reperfusion injury seen in various human gastrointestinal disorders.
Reviewer 2 Report
Comments and Suggestions for Authors
The manuscript "Larazotide acetate protects the intestinal mucosal barrier from anoxia/reoxygenation injury via various cellular mechanisms" aims to highlight the mechanisms that regulate the function of LA in protecting the intestinal mucosa in hypoxic states used to simulate conditions similar to those found in “leaky gut” condition. The manuscript is clear, concise, and well-structured throughout; the data presented are clear and well-explained in the various sections. In my opinion, the paper is worthy of publication in Biomedicines. I suggest just two changes:
- Please reformat the references according to the MDPI style
- Lines 90 – 94. Why were these concentrations chosen? Are they phisiologically relevant? Can they be reached by LA oral administration? Please give information in the text
Author Response
Dear Reviewer,
We sincerely thank you for the thoughtful and constructive comments on our manuscript. We have revised the paper accordingly, improving clarity, methodological detail, and clinical context. Below, we provide a detailed, point-by-point response to each reviewer’s comment. Changes have been highlighted in the revised manuscript.
- Comment: Please reformat the references according to the MDPI style.
Response: We have reformatted the references to adhere to the MDPI style guidelines.
- Comment: Lines 90 – 94. Why were these concentrations chosen? Are they physiologically relevant? Can they be reached by LA oral administration? Please give information in the text.
Response: Thank you for this important question. We acknowledge that the 10 mM concentration is higher than typical clinical doses. Our dose-response data, presented in Figure 1B, show a clear, dose-dependent protective effect of LA on the intestinal barrier during anoxic injury, with the 10 mM concentration providing the most significant protection in our model. We have added a sentence to the manuscript to clarify that this concentration was chosen to elicit a maximal response in our in vitro system, which may not directly translate to in vivo clinical dosage.
Reviewer 3 Report
Comments and Suggestions for Authors
Dear authors,
The study is interesting and of good scientific quality. The introduction is satisfactory. The methods are clear and appropriate for the study. However, some points need to be better presented and clarified.
- In Figure 1B, the effect of anoxia itself does not show a significant impact. Therefore, what is the purpose of using LA without an initial effect?
- What are the isolated effects of LA on normal cells? These controls are not shown and should be included in the study.
- The images in Figure 2B do not allow for good visualization of the results. I suggest reviewing the images and presenting quadrant markings for better identification.
- The discussion is very short and needs to be revised for greater depth.
- The study's perspectives and potential clinical application should be included.
Author Response
Dear Reviewers,
We sincerely thank you for the thoughtful and constructive comments on our manuscript. We have revised the paper accordingly, improving clarity, methodological detail, and clinical context. Below, we provide a detailed, point-by-point response to each reviewer’s comment. Changes have been highlighted in the revised manuscript.
- Comment: In Figure 1B, the effect of anoxia itself does not show a significant impact. Therefore, what is the purpose of using LA without an initial effect?
Response: We appreciate the reviewer's observation. The anoxic injury induced a significant (p<0.05) reduction in TEER, as shown in Figure 1B and described in the results section 3.1: “Anoxic injury resulted in diminished TEER values compared with control cells (p < 0.05), which is indicative of impaired barrier function.” This effect was also previously demonstrated in our another publication: Jin, Y.; Blikslager, A.T. Myosin light chain kinase mediates intestinal barrier dysfunction via occludin endocytosis during anoxia/reoxygenation injury. American Journal of Physiology. Cell Physiology 2016, 311, C996-C1004, doi:10.1152/ajpcell.00113.2016.
- Comment: What are the isolated effects of LA on normal cells? These controls are not shown and should be included in the study.
Response: We thank the reviewer for pointing out this omission. We did perform a similar control in the IPEC-J2 model (Fig. 1C). It increased TEER without the anoxic injury in the IPEC-J2 model. We are more focused on their mechanism of action in this manuscript and suggested some novel mechanisms (e.g., ROCK inhibition) and cellular functions (enhanced proliferation).
- Comment: The images in Figure 2B do not allow for good visualization of the results. I suggest reviewing the images and presenting quadrant markings for better identification.
Response: We agree with the reviewer that the clarity of the images in Figure 2B can be improved. We have added quadrant markings as suggested to more clearly delineate the cell populations and support our conclusions.
4 & 5. Comment: The discussion is very short and needs to be revised for greater depth. The study's perspectives and potential clinical application should be included.
Response: Thank you for this valuable feedback. We agree that the discussion can be significantly strengthened. We have substantially revised and expanded the discussion section to provide a more in-depth analysis of our findings. This new version now includes a detailed section on the potential clinical applications of our results in the context of ischemia-reperfusion injuries and a forward-looking perspective on future research directions needed to translate these findings.
Reviewer 4 Report
Comments and Suggestions for Authors
The paper under review presents the beneficial effects of Larazotide acetate, a synthetic octapeptide, in “leaky gut” by decreasing intestinal permeability and regulating tight junctions. Authors assessed the epithelial barrier protective function of Larazotide in an anoxia/reoxygenation injury in human intestine Caco-2BBe1 monolayers and the leaky model in intestinal porcine epithelial cell-jejunum 2 (IPEC-J2) cell lines, in an effort to reveal the underlined cellular mechanisms.
I have two comments:
- the anoxia/reoxygenation injury model in human intestine Caco-2BBe1 cell line as well as the leaky gut model in swine jejunum epithelial cell lines are first time used or are established and validated models. In the first case, I would like to have more details; otherwise I need please references
- Line 93: Larazotide was applied apically on the Caco-2BBe1 monolayer before anoxia/reoxygenation injury; that means prophylaxis and not treatment. If it is correct, the title is OK, "protects the intestinal mucosal barrier". However, on line 28 "a therapeutic mechanism", line 287 "LA treatment", line 299 "treatment with LA", line 327 "treatment of LA", line 340 "treatment with LA" should be changed accordingly [pre-treatment or something similar]. I also suggest to be included a comment - study limitation on this.
Author Response
Dear Reviewer,
We sincerely thank you for the thoughtful and constructive comments on our manuscript. We have revised the paper accordingly, improving clarity, methodological detail, and clinical context. Below, we provide a detailed, point-by-point response to each reviewer’s comment. Changes have been highlighted in the revised manuscript.
- Comment: the anoxia/reoxygenation injury model in human intestine Caco-2BBe1 cell line as well as the leaky gut model in swine jejunum epithelial cell lines are first time used or are established and validated models. In the first case, I would like to have more details; otherwise I need please references
Response: Thank you for this valuable feedback. The anoxia/reoxygenation injury model in human intestine Caco-2BBe1 cell line was already validated in another of our publications: Jin, Y.; Blikslager, A.T. Myosin light chain kinase mediates intestinal barrier dysfunction via occludin endocytosis during anoxia/reoxygenation injury. American Journal of Physiology. Cell Physiology 2016, 311, C996-C1004, doi:10.1152/ajpcell.00113.2016.
The normal piglet jejunum IPEC-J2 cell model, cultured with intestinal organoids medium, is a novel model we recently developed. You can find more details in the suppl. Fig. 1. Although it has some increase in the different cell types, I agree it cannot recapitulate the complexity of the in vivo intestinal mucosa. We have added a paragraph to the discussion section acknowledging the limitations of using a Caco-2 and IPEC-J2 monoculture. We explicitly state that this model does not include other crucial cell types, such as goblet cells or Paneth cells; therefore, our results should be interpreted within this context.
- Comment: Line 93: Larazotide was applied apically on the Caco-2BBe1 monolayer before anoxia/reoxygenation injury; that means prophylaxis and not treatment. If it is correct, the title is OK, "protects the intestinal mucosal barrier". However, on line 28 "a therapeutic mechanism", line 287 "LA treatment", line 299 "treatment with LA", line 327 "treatment of LA", line 340 "treatment with LA" should be changed accordingly [pre-treatment or something similar]. I also suggest to be included a comment - study limitation on this.
Response: We sincerely thank the reviewer for this crucial and accurate observation. The reviewer is correct that our experimental design evaluates the prophylactic (protective) effect of Larazotide Acetate rather than its therapeutic effect, as it was administered before the injury. We have carefully revised the manuscript to correct this terminology, changing instances of "treatment" to "pre-treatment," "pre-incubation," or "application" where appropriate to accurately reflect our methods. Furthermore, we have added a statement to the discussion acknowledging this as a limitation of the current study.
Reviewer 5 Report
Comments and Suggestions for Authors
Biomedicines 8-13-2025
Here the authors reported that Larazotide acetate (LA), a synthetic octapeptide under development as a potential treatment for celiac disease, protected the intestinal mucosal barrier from anoxia/reoxygenation injury using cell line models. RNA-seq and bioinformatics analysis identified phosphorylation of myosin light chain-2 could be a critical target of LA, which was further studied by the authors. Overall, this is an important study, and some of the results are convincing; however, this Reviewer also raised the following concerns:
Title: “various cellular mechanisms” is too vague, needs to be more specific.
Methods:
“Leaky IPEC-J2 model”: Please provide a reference to this model.
Proliferation and migration assays: Provide the seeding cell numbers.
Figure 2: In 2A, how the cell membrane fraction was isolated needs to be provided in the Methods. Since beta-actin is little expressed in the membrane fraction, a known membrane protein shall be used for quantification.
Figure 3: In the figure legend, “Differential” instead of “Different”
Figure 7: Both the cell proliferation and wound healing assays were performed for a longer period compared to the other studies. What is the half-life of the peptide? It is also difficult to predict whether the observed effect will be beneficial or detrimental in the long term.
Discussion: Please add the information such as how Larazotide acetate gets internalized into the cell, the safety data, etc.
Author Response
- Title: “various cellular mechanisms” is too vague, needs to be more specific.
Response: We agree with the reviewer that the title could be more descriptive of our key findings. We have revised the title to better reflect the specific mechanisms investigated in the manuscript. “Larazotide Acetate Protects the Intestinal Barrier from Anoxia/Reoxygenation Injury via Regulation of Myosin Light Chain Signaling and Epithelial Proliferation.”
model system by measuring the hypoxia inducible factor-1a (HIF-1a) in the previously published article (Jin, Y.; Blikslager, A.T. Myosin light chain kinase mediates intestinal barrier dysfunction via occludin endocytosis during anoxia/reoxygenation injury. American journal of physiology. Cell physiology 2016, 311, C996-C1004, doi:10.1152/ajpcell.00113.2016.).
- Methods: “Leaky IPEC-J2 model”: Please provide a reference to this model.
Response: The normal piglet jejunum IPEC-J2 cell model, cultured with intestinal organoids medium, is a novel model we recently developed. You can find more details in the suppl. Fig. 1. Although it has some increase in the different cell types, it cannot recapitulate the complexity of the in vivo intestinal mucosa. We have added a paragraph to the discussion section acknowledging the limitations of using a Caco-2 and IPEC-J2 monoculture. We explicitly state that this model does not include other crucial cell types, such as goblet cells or Paneth cells; therefore, our results should be interpreted within this context.
- Methods: Proliferation and migration assays: Provide the seeding cell numbers.
Response: We have amended the methods section to include the specific cell seeding densities for these assays.
- Figure 2: In 2A, how the cell membrane fraction was isolated needs to be provided in the Methods. Since beta-actin is little expressed in the membrane fraction, a known membrane protein shall be used for quantification.
Response: We thank the reviewer for these valuable technical suggestions. We have added more details about the membrane fraction isolation method. Regarding the loading buffer, we appreciate your suggestion; however, the samples are no longer available, and we lack the resources and manpower to perform additional work. If it is still considered critical, we will remove Fig. 2A.
- Figure 3: In the figure legend, “Differential” instead of “Different”
Response: We thank the reviewer for catching this typographical error. The legend for Figure 3 has been corrected from "Different" to "Differential."
- Figure 7: Both the cell proliferation and wound healing assays were performed for a longer period compared to the other studies. What is the half-life of the peptide? It is also difficult to predict whether the observed effect will be beneficial or detrimental in the long term.
- Response: This is an excellent point. The half-life of larazotide acetate varies significantly depending on whether it is exposed to enzymes in the GI tract. We didn’t have a specific half-life for our model. To ensure its effects, we kept adding the LA to the media every other day during the proliferation experiments. We also expanded our discussion to consider how the increased proliferation could be either beneficial or detrimental in the long term.
- Discussion: Please add the information such as how Larazotide acetate gets internalized into the cell, the safety data, etc.
Response: We have expanded the discussion to include the proposed mechanism of action and the well-established safety profile of Larazotide Acetate to provide better context for our findings. The lack of information on how LA is internalized into cells and the absence of detectable plasma levels of LA after oral administration are notable. Regarding safety, Larazotide Acetate has been extensively studied in Phase 1, 2, and 3 clinical trials for celiac disease.
Round 2
Reviewer 1 Report
Comments and Suggestions for Authors
Well done. The authors have answered all my questions. Thank you!
Author Response
I appreciate your review!
Reviewer 3 Report
Comments and Suggestions for Authors
Dear author.
The new version is better. In this sense, I suggest that a minor revision will be made. The graphics need better quality. I suggest that you make a reorganisation. In the statistical analysis, please organise the "P" value, because in the figures you show "***", but in the description you present only P<0.05. Normalise these values.
Best regards.
Author Response
Dear reviewer,
Thank you for your suggestion. I noticed the low resolutions of some figures and have updated them with higher resolutions. The "P" values have been revised throughout the manuscript as you suggested.
Reviewer 5 Report
Comments and Suggestions for Authors
The authors have addressed the concerns.
Author Response
Thank you!